# WHEN HARD NEGATIVE SAMPLING MEETS SUPER-VISED CONTRASTIVE LEARNING

## ABSTRACT

State-of-the-art pre-trained image models predominantly follow a two-stage strategy: pre-training on large datasets and fine-tuning on a task-specific labeled dataset with a cross-entropy objective function. However, many studies have shown that cross-entropy can result in sub-optimal generalization and stability. While supervised contrastive learning addresses some limitations of cross-entropy objective function by emphasizing intra-class similarities and inter-class differences, it neglects the importance of hard negative mining. We hypothesize that weighting negative samples by their dissimilarity to positives enhances the efficacy of contrastive learning. This paper introduces a new supervised contrastive learning objective function, named SCHaNe, which incorporates hard negative sampling during the fine-tuning phase. Without requiring specialized architectures, additional data, or extra computational resources, experimental results indicate that SCHaNe outperforms the strong baseline BEiT-3 in Top-1 accuracy across twelve benchmarks, with significant gains of up to 3.32% in few-shot learning settings and 3.41% in full-dataset fine-tuning. Importantly, our proposed objective function sets a new state-of-the-art for base models (parameter size at 88 million) on ImageNet-1k, achieving an accuracy of 86.14%. Furthermore, we demonstrate that the proposed objective function yields better embeddings and explains the improved effectiveness observed in our experiments. Our code is available at `https://anonymous.4open.science/r/SCHaNe-61C6/`.

## 1 INTRODUCTION

Approaches for achieving state-of-the-art performance in image classification tasks often employ models that are initially pre-trained on auxiliary tasks and then fine-tuned on a task-specific labeled dataset with a cross-entropy objective function (CE) (Dosovitskiy et al., 2021; Wang et al., 2022; He et al., 2022). This prevalent approach, however, leads to sub-optimal results due to the limitations of CE. Specifically, the measure of KL-divergence between one-hot label vectors and model outputs can hinder generalization (Liu et al., 2016; Cao et al., 2019) and has been shown to be sensitive to noisy labels (Nar et al., 2019; Liu et al., 2016) or adversarial samples (Elsayed et al., 2018; Nar et al., 2019). Various techniques have emerged to address these problems, such as knowledge distillation (Hinton et al., 2015), self-training (Yalniz et al., 2019), Mixup (Zhang et al., 2017), CutMix (Yun et al., 2019), and label smoothing (Szegedy et al., 2015). However, in scenarios such as few-shot learning, the issues with CE have not been fully mitigated. Indeed, techniques such as extended fine-tuning epochs and specialized optimizers (Zhang et al., 2020; Mosbach et al., 2020) can reduce the impact of cross-entropy to some extent, but introduce new challenges, such as extended training durations and increased complexity.

Amidst these challenges, contrastive learning emerges as a potential solution, notably in few-shot learning approaches such as SimCLR (Chen et al., 2020). To fully exploit the potential of contrastive learning, it is crucial to adhere to its foundational principles: promoting similarity among positive pairs (intra-class data points) and maximizing differences for negative pairs (inter-class data points). Task-specific label information becomes indispensable in correct identification of positive pairs. Building upon such foundations, there is a growing interest in combining task labels with contrastive learning, seeking to improve vanilla cross-entropy pre-training approaches (Russakovsky et al., 2015). A crucial observation is that numerous state-of-the-art supervised contrastive learning methods (Russakovsky et al., 2015; Gunel et al., 2020) as well as unsupervised approaches (Chen

et al., 2020; Sohn, 2016; Ge et al., 2018) neglect the nuanced selection of negative samples, treating them without differentiation or prioritization during selection or processing. This generalization misses out on the demonstrated benefits of leveraging "hard" negative samples for accelerated corrective learning, as underscored by several studies (Song et al., 2015; Schroff et al., 2015). Motivated by these successes and research gaps, we ask: *Can explicit hard negative mining improve supervised contrastive learning, particularly during the fine-tuning phase, and address the shortcomings of cross-entropy objective function?*

In response to this evident research gap, we introduce a novel supervised contrastive learning objective function called SCHaNe. This approach combines supervised contrastive learning with hard negative sampling to optimize the selection of positive and negative samples, thereby achieving state-of-the-art performance. Empirical validation, as highlighted in Figure 1, shows that our SCHaNe places greater emphasis on negative samples that are challenging to differentiate from positive samples. These challenging samples are typically closer in the feature space to the positive samples. By doing so, SCHaNe encourages the encoder to produce more distinguishable embeddings, consequently improving performance in downstream tasks such as classification. Our main contributions are:

- **Introduction of a Novel Learning Objective Function**: We propose the SCHaNe contrastive learning objective function that results in more distinguishable embeddings by prioritizing hard negative samples. SCHaNe enhances model performance without requiring specialized architectures or additional resources. To the best of our knowledge, this is the first work to integrate explicit hard negative mining with a supervised contrastive objective function.

- **Superior Performance in Few-Shot Learning**: When benchmarked against the very strong BEiT-3 baseline, our approach significantly improves Top-1 accuracy across four few-shot datasets. This improvement is especially evident in challenging 1-shot learning scenarios. Specifically, we observe a boost of up to 3.32% in Top-1 accuracy on the CIFAR-FS (Bertinetto et al., 2019) dataset compared to the BEiT-3 baseline.

- **State-of-the-Art in Full Dataset Fine-Tuning**: Our method consistently outperforms competitors across eight image datasets, establishing a new state-of-the-art for base models on ImageNet-1k. Furthermore, our approach achieves a notable increase of 3.41% on the iNaturalist 2017 dataset (Horn et al., 2018) compared to the BEiT-3 baseline.

## 2 RELATED WORK

### 2.1 LIMITATIONS OF CROSS-ENTROPY OBJECTIVE FUNCTION

The cross-entropy objective function (CE) has long been the default setting for many deep neural models due to its ability to optimize classification tasks effectively. However, recent research has revealed several inherent drawbacks (Liu et al., 2016; Cao et al., 2019). Specifically, models trained with the CE tend to exhibit poor generalization capabilities. This vulnerability stems from the model having narrow decision margins in the feature space, making it more susceptible to errors introduced by noisy labels (Nar et al., 2019; Liu et al., 2016) and adversarial examples (Elsayed et al., 2018; Nar et al., 2019). These deficiencies underscore the need for alternative objective functions that offer better robustness and discrimination capabilities.

### 2.2 CONTRASTIVE LEARNING

Contrastive learning, originally introduced by Becker & Hinton (1992), aims to improve the distinction between samples by minimizing the distance between positive pairs (instances of the same class) and maximizing the distance for negative pairs (instances of different classes). This approach has been transformative, especially in the realm of computer vision. For example, SimCLR (Chen et al., 2020) leveraged instance-level comparisons for image classification in an unsupervised manner, setting performance benchmarks on datasets such as ImageNet ILSVRC-2012 (Russakovsky et al., 2015). Furthermore, adopting label information, as explored by Khosla et al. (2020), has propelled contrastive learning to a new level, revealing it as a viable, if not superior, alternative to cross-entropy objective function. This advancement underscores the robustness of the supervised contrastive objective function when subjected to a range of hyperparameter settings and data augmentation techniques.

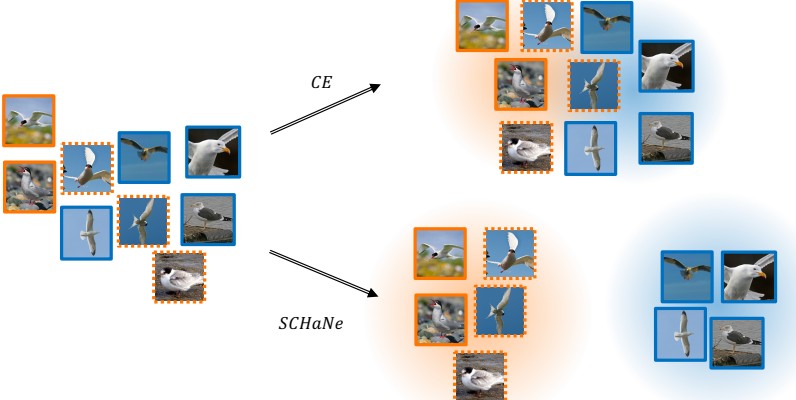

Figure 1: SCHaNe, our proposed objective function, integrates Supervised Contrastive Learning with Hard Negative sampling and a Cross-Entropy (CE) term. Illustrated with CUB-200-2011 samples, it emphasizes hard negatives (thick dashed borders) for better class separation. This underscores their marked visual similarity to their positive counterparts. On the left, blue indicates positive examples and orange denotes negatives. On the right, SCHaNe's fine-tuning visibly separates class embeddings more effectively than vanilla CE.

## 2.3 THE CRITICALITY OF NEGATIVE MINING

While the benefits of positive sample identification are well-established, the nuanced role of negative samples, particularly hard negatives, has been comparatively underexplored. Some pioneering works, such as Chuang et al. (2020), address the challenge of identifying true negatives from a large pool of potential negatives by employing unlabeled positive learning principles to approximate the true negative distribution. The integration of mixup techniques Zhang et al. (2017) with contrastive learning, as proposed by Kalantidis et al. (2020), further emphasizes the potential of mining hard negatives in the latent space. Numerous studies (Song et al., 2015; Kumar et al., 2017; Wu et al., 2017; Ge et al., 2018; Suh et al., 2019) have consistently underlined the critical role of these hard negatives in refining the discriminative power of embeddings.

Building upon these advancements, our work seeks to integrate the advantages of supervised contrastive learning with the precision of hard negative sampling. We propose an innovative SCHaNe objective function that not only recognizes the importance of positive samples but also accentuates the criticality of effectively weighting negative samples, particularly hard negatives, for improving model performance.

## 3 APPROACH

We propose a novel supervised contrastive learning objective function (SCHaNe) for fine-tuning pre-trained image models that integrates Supervised Contrastive objective function with the Hard Negative sampling (SCLN) and the Cross-Entropy objective function (CE). The aim is to encapsulate similarities between instances of the same class (positive samples) using label information and to contrast them against instances from other classes (negative samples), with particular emphasis on hard negative samples. SCHaNe's foundation draws from the premise that the training efficacy of negative samples varies between soft and hard samples. We argue that weighting negative samples based on their dissimilarity to positive samples is more effective than treating them equally. This allows the model to prioritize distinguishing between positive samples and those negative samples that the embedding deems similar to the positive ones, ultimately enhancing overall performance.

## 3.1 SCHaNe: SUPERVISED CONTRASTIVE LEARNING WITH HARD NEGATIVE SAMPLING

The overall proposed objective function is a weighted combination of Supervised Contrastive objective function with Hard Negative sampling (SCLN) and vanilla Cross-Entropy objective function (CE), as expressed in Equation 1:

$$\mathcal{L}_{\text{SCHaNe}} = (1 - \lambda)\mathcal{L}_{\text{CE}} + \lambda\mathcal{L}_{\text{SCLN}} \tag{1}$$

In Equation 1, the term $\mathcal{L}_{\text{CE}}$ represents the Cross-Entropy objective function, while $\mathcal{L}_{\text{SCLN}}$ symbolizes our proposed supervised contrastive objective function. $\lambda$ represents a scalar weighting hyperparameter tuned for each specific task and setting. $\lambda$ determines the relative importance of each of the two losses. If $\lambda$ is 1, only the SCLN objective function is considered; if it is 0, only the Cross-Entropy objective function is considered. Any value in between gives a mixture of the two.

To provide context for $\mathcal{L}_{\text{CE}}$, we refer to the vanilla definition of the multi-class Cross-Entropy objective function, detailed in Equation 2:

$$\mathcal{L}_{\text{CE}} = -\frac{1}{N}\sum_{i=1}^{N}\sum_{c=1}^{C} z_{i,c}\log(\hat{z}_{i,c}) \tag{2}$$

In Equation 2, $z_{i,c}$ and $\hat{z}_{i,c}$ represent the label and the model's output probability for the $i$th instance belonging to class $c$, respectively. We then introduce $\beta$, a weight factor for each negative sample, as described in Equation 3:

$$\beta = \exp(z_i \cdot z_k/\tau)\frac{|\mathcal{D}^{*\text{-}}_{-z_i}|}{\displaystyle\sum_{z_k \in \mathcal{D}^{*\text{-}}_{-z_i}}\exp(z_i \cdot z_k/\tau)} \tag{3}$$

The weight is based on the embeddings' dot product (or similarity) and is normalized by a temperature parameter $\tau$. The purpose of this equation is to give more emphasis to the hard negative samples, i.e. the negatives that the current embeddings find most similar to the positive samples. From Eq. 3, we introduce the formal definition of our SCHaNe objective function, illustrated in Equation 4:

$$\mathcal{L}_{\text{SCLN}} = \sum_{z_i \in \mathcal{D}^*}\log\frac{-1}{|\mathcal{D}^{*+}_{-z_i}|}\frac{\displaystyle\sum_{z_p \in \mathcal{D}^{*+}_{-z_i}}\exp(z_i \cdot z_p/\tau)}{\displaystyle\sum_{z_p \in \mathcal{D}^{*+}_{-z_i}}\exp(z_i \cdot z_p/\tau) + \sum_{z_k \in \mathcal{D}^{*\text{-}}_{-z_i}}\beta\exp(z_i \cdot z_k/\tau)} \tag{4}$$

In Equation 3 and 4, $\mathcal{D}^*$ represents the entire mini-batch composed of an embedding $z$ for each image view (or anchor) $i$. Therefore, $z_i \in \mathcal{D}^*$ is a set of embeddings within the mini-batch. The superscripts $+$ and $-$, e.g. $\mathcal{D}^{*+}$, denote sets of embeddings consisting only of positive and negative examples, respectively, for the current anchor within the mini-batch. The term $|\mathcal{D}^{*+}_{-z_i}|$ represents the cardinality of the positive set for the current anchor, while the subscript $-z_i$ denotes that this set excludes the embedding $z_i$. The symbol $\cdot$ represents the dot product. $\tau$ is a scalar temperature parameter controlling class separation. A lower value for $\tau$ encourages the model to differentiate positive and negative instances more distinctly.

Equation 4 is the core of our contribution. It aims to minimize the distance between the embeddings of positive pairs (the same class) and maximize the distance between the anchor and negative samples, especially the hard negatives. This equation has two main parts: (1) The numerator focuses on the positive samples and aims to make their embeddings close to the anchor embedding. (2) The denominator contains both positive and weighted negative samples. The goal is to ensure that the anchor's embedding is distant from negative samples, especially the hard samples. The weights (or importance) of these negative samples are given by the previously calculated $\beta$ parameter.

## 3.2 Analysis of SCHaNe

To enhance the effectiveness of contrastive learning, our approach is rooted in three foundational principles that have been empirically supported by numerous studies (Chen et al., 2020; Russakovsky et al., 2015; Chuang et al., 2020; Kalantidis et al., 2020):

- **True Positives**: Samples must strictly share the same label as the anchor $x$, which drives the model to discern inherent similarities.

- **True Negatives**: Samples with distinct labels from the anchor $x$ ensure computational focus on genuinely contrasting pairs.
- **Emphasis on Hard Negatives**: The core of effective contrastive learning is the model's ability to discern between closely related samples. Therefore, negative samples that the model's current embedding perceives as akin to the anchor, termed "hard" negative samples, are the most instructive. They push the model's boundaries, facilitating more refined feature extraction.

Reflecting on these principles, our proposed supervised contrastive objective function with hard negative sampling achieves the following:

- **Robust Positive/Negative Differentiation**: We ensure a categorical distinction between true positive and true negative samples by leveraging explicit label information, as encapsulated in Equation 4. This not only prevents the model from being misled by incorrectly labeled samples but also reinforces the core philosophy of contrastive learning. The aim is two-fold: to reduce the distance between the embeddings of positive pairs and to widen the gap for negative pairs, ensuring robust class separation.
- **Discriminating Fine Detail with Hard Negatives**: Our methodology adjusts the weighting of negative samples based on their similarities to positive instances, as defined in Equation 3. This nuanced approach ensures that the model not only differentiates between glaringly distinct samples but also hones its skills on the more challenging, closely related negative samples. Such an approach paves the way for a robust model that discerns in real-world scenarios where differences between classes might be minimal.

This refined approach to contrastive learning promises to introduce models with enhanced discriminative power, fine-tuned to the intricate details of the data on which they are trained.

### 3.3 RELATIONSHIP TO OTHER CONTRASTIVE LEARNING APPROACHES

Contrastive learning has gained significant traction for robust representation learning, notably in computer vision (Chen et al., 2020; He et al., 2019; Tian et al., 2020a; Kolesnikov et al., 2019). Such approaches, which function without labeled data, are particularly appealing for pre-training, when given the vast amount of available unlabeled data. Notable methods such as SimCLR (Chen et al., 2020) achieve competitive performance even without labels, although fully outperforming supervised methods remains challenging. This unsupervised strategy raises questions about the effectiveness of these contrastive learning frameworks, especially during fine-tuning. With many downstream datasets having labels, ignoring them limits the achievable performance. For instance, Russakovsky et al. (2015) proposed a supervised contrastive objective function that exploits label information during pre-training. Still, Russakovsky et al. (2015) relies on implicit hard negative mining, which exhibits limitations. In the realm of natural language processing, Gunel et al. (2020) effectively applied supervised contrastive objective function during fine-tuning but also did not leverage label data for hard negative mining. This leaves a research gap: exploring the effectiveness of supervised contrastive learning with hard negative sampling, especially in the context of fine-tuning in the vision domain.

### 3.4 REPRESENTATION LEARNING FRAMEWORK

We introduce a representation learning framework comprised of three main components, designed specifically to optimize our SCHaNe objective function:

- **Data Augmentation module,** $Aug(\cdot)$: This component creates two different views of each sample $x$, denoted $\tilde{x} = Aug(x)$. This means that every sample will have at least one similar sample (positive pair) in a batch during fine-tuning.
- **Encoder Network,** $Enc(\cdot)$: This network encodes the input data, $x$, into a representation vector, $r = Enc(x)$. Each of the two different views of the data is fed into the encoder separately.
- **Classification head,** $Head(\cdot)$: This maps the representation vector, $r$, to probabilities of classes in the target task. The mapping primarily consists of a linear layer, and we utilize its output to calculate the cross-entropy objective function.

| | | CIFAR-FS | | FC100 | | miniImageNet | | tieredImageNet | |
|---|---|---|---|---|---|---|---|---|---|
| Model | Method | 1-shot | 5-shot | 1-shot | 5-shot | 1-shot | 5-shot | 1-shot | 5-shot |
| Dhillon et al. (2020) | Transductive | 76.58±0.68 | 85.79±0.50 | 43.16±0.59 | 57.57±0.55 | 65.73±0.68 | 78.40±0.52 | 73.34±0.71 | 85.50±0.50 |
| Zhang et al. (2021) | Meta-QDA | 75.83±0.88 | 88.79±0.75 | - | - | 67.83±0.64 | 84.28±0.69 | 74.33±0.65 | 89.56±0.79 |
| Hiller et al. (2022) | FewTRUE-ViT | 76.10±0.88 | 96.14±0.64 | 46.20±0.79 | 63.14±0.73 | 68.02±0.88 | 84.51±0.53 | 72.96±0.92 | 87.79±0.67 |
| Hiller et al. (2022) | FewTRUE-Swin | 77.76±0.81 | 88.90±0.59 | 47.68±0.78 | 63.81±0.75 | 72.40±0.78 | 86.38±0.49 | 76.32±0.87 | 89.96±0.55 |
| Hu et al. (2023) | BAVARDAGE | 82.68±0.25 | 89.97±0.18 | 52.60±0.32 | 65.35±0.25 | 77.85±0.28 | 88.02±0.14 | 79.38±0.29 | 88.04±0.18 |
| BEiT-3 | CE | 83.68±0.80 | 93.01±0.38 | 66.35±0.95 | 84.33±0.54 | 90.62±0.60 | 95.77±0.28 | 84.84±0.70 | 94.81±0.34 |
| BEiT-3 | SCHaNe | **87.00±0.70** | **93.77±0.36** | **69.87±0.91** | **87.06±0.52** | **92.35±0.53** | **96.78±0.23** | **87.24±0.62** | **96.09±0.29** |

Table 1: **Comparison to baselines on the few-shot learning setting.** Average few-shot classification accuracies(%) with 95% confidence intervals on test splits of four few-shot learning datasets.

| Model | Method | CIFAR-100 | CUB-200-2011 | Caltech-256 | Oxford 102 Flowers | Oxford-IIIT Pet | iNat2017 | Places365 | ImageNet-1k |
|---|---|---|---|---|---|---|---|---|---|
| ViT-B | CE | 87.13 | 76.93 | 90.92 | 90.86 | 93.81 | 65.26 | 54.06 | 77.91 |
| MAE | CE | 87.67 | 78.46 | 91.82 | 91.67 | 94.05 | 70.50 | 57.90 | 83.60 |
| BEiT-3 | CE | 92.96 | 98.00 | 98.53 | 94.94 | 94.49 | 72.31 | 59.81 | 85.40 |
| BEiT-3 | SCHaNe | **93.56** | **98.93** | **99.41** | **95.43** | **95.62** | **75.72** | **62.22** | **86.14** |

Table 2: **Comparison to baselines on full dataset fine-tuning setting.** Classification accuracies on eight various datasets.

Our SCHaNe objective function (Eq. 4) can be applied using a wide range of encoders, such as BEiT-3 (Wang et al., 2022) for computer vision applications and models such as BERT (Devlin et al., 2019) for natural language processing tasks. In this paper, our primary focus is on fine-tuning pre-trained image models. Following the method in Chen et al. (2020), every image in a batch is altered to produce two separate views (anchors). Views with the same label as the anchor are considered positive, while the rest are viewed as negative. The encoder output, represented by $y_i = Enc(z_i)$, is used to calculate the contrastive objective function. In contrast, the output from the classification head, denoted as $p_i = Head(Enc(z_i))$, is used for the cross-entropy objective function. We have incorporated L2 normalization on encoder outputs, a strategy demonstrated to enhance performance significantly (Tian et al., 2020b).

## 4 EVALUATION

We evaluate our proposed objective function in two primary image classification scenarios: few-shot learning and full-dataset fine-tuning.

**Few-shot Learning.** Four prominent benchmarks are used for evaluation: CIFAR-FS (Bertinetto et al., 2019), FC100 (Oreshkin et al., 2018), miniImageNet (Vinyals et al., 2016), and tieredImageNet (Ren et al., 2018). We follow established splitting protocols for a fair comparison (Bertinetto et al., 2019; Oreshkin et al., 2018; Ravi & Larochelle, 2017). Each test run comprises 3,000 randomly sampled tasks, and we report median Top-1 accuracy with a 95% confidence interval across three runs, maintaining a consistent query shot count of 15.

**Full-dataset Fine-tuning.** Our evaluation leverage eight renowned datasets: CIFAR-100 (Krizhevsky et al., 2009), CUB-200-2011 (Wah et al., 2011), Caltech-256 (Griffin et al., 2007), Oxford 102 Flower (Nilsback & Zisserman, 2008), Oxford-IIIT Pet (Parkhi et al., 2012), iNaturalist 2017 (Horn et al., 2018), Places365 (Zhou et al., 2014), and ImageNet-1k (Deng et al., 2009). We adhere to official train/test splits and report mean Top-1 test accuracy over three different initializations.

**Implementation Details.** We utilize the base model of BEiT-3 (Wang et al., 2022) as our encoder backbone due to its state-of-the-art performance on ImageNet-1k. For SCHaNe experiments, a grid-based hyperparameter search is conducted on the validation set. Optimal settings ($\tau = 0.5$ and $\lambda = 0.9$) are employed because they consistently yield the highest validation accuracies. More details are in Appendix A.2.

### 4.1 EXPERIMENTAL RESULTS

**Few-shot Results.** Table 1 shows the few-shot learning results. This evaluation emphasizes the model's ability to generalize well with limited labeled data for each class. Our BEiT-3-CE baseline (fine-tuned with vanilla cross-entropy) already outperforms all other baselines by up to $26.76\%$ in Top-1 accuracy. The proposed SCHaNe brings further consistent and significant improvement to BEiT-3 (denoted BEiT-3-SCHaNe) in Top-1 accuracy across the four selected few-shot datasets over the BEiT-3-CE baseline. It is worth noting that there is a statistically significant improvement when comparing the results of BEiT-3-SCHaNe with BEiT-3-CE across the four few-shot datasets, using the paired t-test with $p < 0.01$. In particular, in the 1-shot learning setting, the most significant improvement is observed on FC100 in comparison to the BEiT-3 baseline, with an accuracy increase

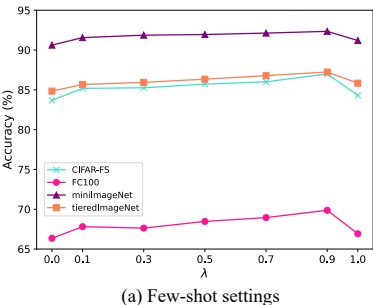
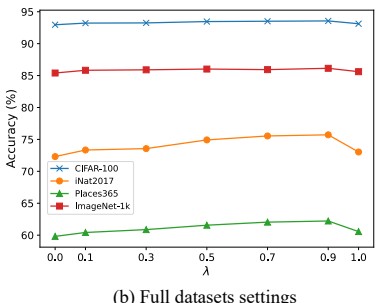

(a) Few-shot settings

(b) Full datasets settings

Figure 2: **Evaluation of the impact of the $\lambda$ hyperparameter.** Results on eight tested datasets with $\lambda$ values ranging from $\{0, 0.1, 0.3, 0.5, 0.7, 0.9, 1.0\}$.

of $3.32\%$ from $66.35\%$ to $69.87\%$. On average, there is an increase of $2.7\%$ in accuracy across all datasets. For the 5-shot learning setting, the improvements are again notable. We observe the largest increase from $84.33\%$ to $87.06\%$ on FC100, and on average, an increase of $1.4\%$ in accuracy is observed across four tested datasets. The performance difference between 1-shot and 5-shot learning indicates that our proposed SCHaNe is more effective as the number of positive samples for each class decreases. Additionally, there is reduced variability in the results, as evidenced by tighter confidence intervals for Top-1 accuracy compared to BEiT-3-CE. Overall, these outcomes underline the efficacy of our proposed Supervised Contrastive Objective Function with Hard Negative sampling (SCHaNe), refining BEiT-3's pre-trained representations for few-shot learning tasks.

**Full-dataset Results.** Table 2 presents the results of full-dataset fine-tuning, which offers further evidence of the applicability of our proposed SCHaNe beyond few-shot scenarios. Our proposed SCHaNe consistently boosts classification accuracy across eight image classification datasets when compared to three state-of-the-art image models with vanilla cross-entropy fine-tuning. We observe a statistically significant improvement when comparing the BEiT-3-SCHaNe results with BEiT-3-CE on the tested datasets, using the paired t-test with $p < 0.05$. In some instances, the increase in performance by our SCHaNe (BEiT-3-SCHaNe) is modest, given the particularly strong baseline already established by BEiT-3-CE, as seen with a modest rise in accuracy, from $98.00\%$ to $98.93\%$ on CUB-200-2011. However, for more challenging datasets, such as iNaturalist2017, we observe a significant increase in classification accuracy, rising from $72.31\%$ to $75.72\%$. We argue that this larger degree of performance gain is due to the dataset's extensive class variety, e.g., 5087 classes in iNaturalist2017 vs. 200 classes in CUB-200-2011. A similar increase is evident in ImageNet-1k, where the accuracy increased from $85.40\%$ to $86.14\%$. It is important to highlight that, at the time of writing, BEiT-3-SCHaNe achieves a new state-of-the-art accuracy of $86.14\%$ for base models (with parameters of around 87 million) on the leaderboard on ImageNet [1]. In essence, our results emphasize the prowess of the SCHaNe in amplifying the performance of pre-trained models during subsequent fine-tuning across a range of datasets. Moreover, our approach is intuitive and achieves these results without the need for specialized architectures, extra data, or increased computational overhead, making it a compelling and powerful alternative to the vanilla cross-entropy objective function commonly used.

## 5 RESULT ANALYSIS

### 5.1 OPTIMIZING $\lambda$: BRIDGING CROSS-ENTROPY AND CONTRASTIVE LEARNING

Our proposed SCHaNe incorporates a hyperparameter, $\lambda$, to control the contributions of the Cross-Entropy objective function (CE) term and the proposed Supervised Contrastive objective function with Hard Negative sampling (SCLN) term, as shown in Equation 1. Specifically, for $\lambda = 0$, the objective function becomes equivalent to the vanilla CE, while for $\lambda = 1$, the objective function becomes exclusively the SCLN term. To understand the influence of $\lambda$, we evaluate its effect on classification accuracy in a one-shot setting across four few-shot learning datasets. Figure 2 presents the test accuracy for varying values of $\lambda$.

Our experiments reveal a consistent trend: as the weight assigned to the SCLN term ($\lambda$) increases, performance progressively improves across all tested datasets, peaking at $\lambda = 0.9$. For instance,

---

[1]https://paperswithcode.com/sota/image-classification-on-imagenet

| Model | Method | Label | HNS | Acc(%) |
|-------|--------|-------|-----|--------|
| BEiT-3 | CE | ✓ | | 83.68 |
| BEiT-3 | CE + SimCLR | | | 73.99 |
| BEiT-3 | CE + SupCon | ✓ | | 85.32 |
| BEiT-3 | SCHaNe | ✓ | ✓ | 87.00 |

Table 3: **Ablation study on CIFAR-FS.** Results are reported in the 1-shot few-shot learning setting using Top-1 accuracy. "CE" denotes cross-entropy objective function, "Label" represents task labels, and "HNS" refers to hard negative sampling.

this optimal setting yields an average performance boost of $2.14\%$ and $2.74\%$ over the exclusive use of either the SCLN or CE term on four few-shot datasets. Nevertheless, eliminating the CE term ($\lambda = 1$) results in a performance drop, e.g., an accuracy drop from $87.0\%$ to $84.3\%$ on CIFAR-FS. This trend also manifests in full-dataset fine-tuning, highlighting the complementary nature of CE and SCLN. Thus, optimizing this balance is crucial for maximizing performance with SCHaNe.

## 5.2 ABLATION STUDY ON THE IMPACT OF LABELS AND NEGATIVE SAMPLES

We aim to verify that the performance enhancements resulting from our proposed objective function do not simply stem from the incorporation of a contrastive learning term. Thus, we conduct an ablation study to delve deeper into this. Specifically, we evaluate two alternative contrastive objective functions: a supervised contrastive objective function without the hard negative weighting (SupCon), and an unsupervised version of SupCon (SimCLR). In the same way as for SCHaNe, we adjust the weighting of the contrastive term using the parameter $\lambda$ with a value of $0.9$ for these variants. We utilize the same BEiT-3 backbone in all experiments and report our findings on the CIFAR-FS dataset.

As Table 3 illustrates, integrating a contrastive learning term indeed increases performance in the fine-tuning stage. Importantly, the inclusion of label information in the SupCon objective function leads to a significant increase in performance of $11.33\%$ in contrast to the unsupervised approach (SimCLR). However, our proposed SCHaNe objective function further improves the classification accuracy from $85.32\%$ to $87.00\%$, surpassing the results obtained with SupCon. These findings support our argument that a supervised contrastive objective function, especially when paired with explicit hard negative sampling, enhances the accuracy during fine-tuning of pre-trained models, particularly in few-shot learning scenarios.

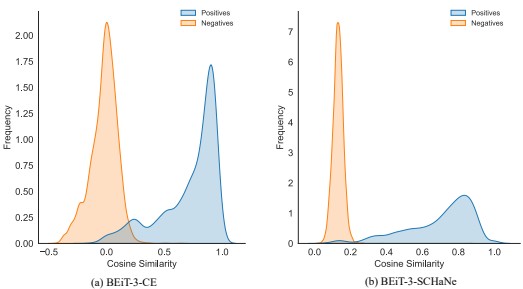

Figure 3: **Plot of cosine similarity distribution across two random classes from CIFAR-100.** Blue represents similarities of positive samples, while orange represents similarities of negative samples.

## 5.3 EMBEDDING QUALITY ANALYSIS: SCHaNe VS. CROSS-ENTROPY

To validate the enhancements brought by the SCHaNe objective function, we perform a thorough evaluation focusing on the geometric characteristics of the generated representation spaces. We hypothesize that our SCHaNe is accentuating the geometric nature of the embedding space, thereby enhancing class distinction and improving transfer learning performance. To elaborate, we examine BEiT-3 fine-tuned with vanilla cross-entropy (denoted as BEiT-3-CE) and compare it to the model when fine-tuned using our SCHaNe (denoted as BEiT-3-SCHaNe). Specifically, we evaluate two key facets of these models:

- **Distributions of cosine similarities between image pairs**: This assessment provides insights into how well the model differentiates between classes in the embedding space.
- **Visualization of the embedding space using the t-SNE algorithm (Van der Maaten & Hinton, 2008)**: This visualization allows us to observe the separation or clustering of data points belonging to different classes, offering a spatial understanding of the embeddings.

In Figure 3, we present the pairwise cosine similarity distributions of BEiT-3-CE and BEiT-3-SCHaNe embeddings in Figure 3. Specifically, we randomly select two classes from CIFAR-100 to compute cosine similarities for positive (same class) and negative pairs (different classes). Observations from these plots reveal that the BEiT-3-SCHaNe embeddings demonstrate superior separation between classes and less overlap between positive and negative samples compared to BEiT-3-CE.

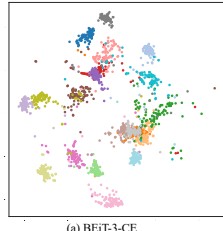 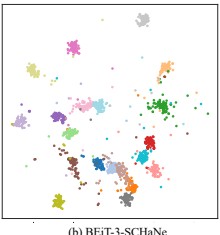

(a) BEiT-3-CE      (b) BEiT-3-SCHaNe

Figure 4: **Embedding Space Visualization for BEiT-3-CE vs. BEiT-3-SCHaNe.** Displayed over twenty CIFAR-100 validation set classes using t-SNE. Each dot represents a sample, with distinct colors indicating different label classes.

In Figure 4, the t-SNE algorithm visualizes the embedding space of BEiT-3-CE and BEiT-3-SCHaNe for twenty CIFAR-100 classes. The BEiT-3-CE embeddings (Fig. 4a) display instances where the same class nodes are closely packed but also reveal several outliers. This suggests a reduced discriminative capability. On the contrary, BEiT-3-SCHaNe embeddings (Fig. 4b) display more separated and compact class clusters, suggesting improved classification capabilities.

In summary, our analysis demonstrates the proposed SCHaNe objective function restructures the embedding space to enhance class distinction, addressing the limitations of the vanilla cross-entropy objective function. This demonstrates the limitations of using cross-entropy objective function and the value-add from integrating the supervised contrastive objective function with a hard negative sampling technique (SCLN in this case) into the fine-tuning stage. This enhancement is particularly effective for few-shot learning scenarios, where limited labelled data requires the model to rely more on high-quality, discriminative representations.

## 6   Conclusion and Future Work

In this work, we first propose a supervised contrastive learning objective function with hard negative sampling, named SCHaNe, to address the shortcomings of Cross-Entropy and existing contrastive learning losses during fine-tuning. By effectively leveraging labels to discern true positives from negatives and emphasizing those hard negative samples based on dissimilarity to positive counterparts, SCHaNe encourages models to generate more distinguishable embeddings. Our experimental results demonstrate statistically significant accuracy improvements across all twelve tested datasets upon a state-of-the-art baseline, BEiT-3, with significant gains of up to 3.32% in few-shot learning settings and 3.41% in full-dataset fine-tuning. Importantly, SCHaNe establishes a new state-of-the-art for base models (parameters of around 87 million) on ImageNet-1k with an accuracy of 86.14%. These achievements confirm that our proposed SCHaNe effectively improves the performance of pre-trained models without requiring specialized architectures or additional resources. Our ablation study further validates our approach by confirming the critical roles of label information and hard negative sampling. To encapsulate, our comprehensive investigations and robust empirical evidence compellingly substantiate our methodological decisions, underscoring that SCHaNe serves as a superior alternative of the cross-entropy objective function for augmenting the performance of pre-trained models in the realm of image classification. Future research will focus on applying SCHaNe in the pre-training phase and extending its applicability to other types of models such as graph neural networks and convolutional neural networks.

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

| Model | iNaturalist2017 | Imagenet-1k | Places365 |
|---|---|---|---|
| BEiT3-CE | 0.3230 | 0.2723 | 0.3404 |
| BEiT3-SCHaNe | 0.9827 | 0.9201 | 0.9338 |

Table 4: **Comparison of Isotropy Score across three datasets for BEiT-3-CE and BEiT-3-SCHaNe.** A higher value is better. A higher Isotropy Score indicates better isotropy and generalizability.

## A APPENDIX

### A.1 REPRODUCIBILITY STATEMENT

The experiments for SCHaNe primarily utilize the BEiT-3 codebase[2]. Detailed experimental setup is elaborated in Section 4, while additional implementation specifics are provided in Appendix A.2. Furthermore, an anonymous link to our complete source code is included in the abstract, and a supplementary copy is also uploaded with the supplemental material.

### A.2 IMPLEMENTATION DETAILS

As transformer models increase in size, they generally exhibit enhanced performance but also incur escalating costs for both training and deployment. To ensure a fair comparison in our experiments, we standardize the model size by selecting the base configuration of BEiT-3, whose architecture is roughly commensurate with the standard "base" models prevalent in existing literature and employed as our baselines. Specifically, our chosen 'base' model comprises 12 transformer layers, 768 hidden units per layer, and 12 attention heads. Moreover, given that this paper focuses on evaluating the efficacy of fine-tuning, we commence by utilizing a pre-trained BEiT-3 model.

We conduct training over 100 epochs for the few-shot datasets and 50 epochs for the full datasets. We apply an early-stopping strategy during training, terminating the training when the validation loss does not decrease for 10 epochs. For testing stage in few-shot settings, we train an N-way logistic regression base classifier. We use the implementations in scikit-learn [3] for the base classifier. For data augmentation, we employ AutoAugment (Cubuk et al., 2019), which has proven to be highly effective for supervised contrastive learning (Russakovsky et al., 2015). Throughout all fine-tuning experiments, we choose the Adam optimizer with a learning rate set at $1 \times 10^{-4}$, a weight decay of 0.05, and a batch size of 1024 (unless specified otherwise). We also integrate a dropout rate of 0.1. In experiments that incorporate the SCHaNe term, we perform an extensive grid-based hyperparameter search on the validation set, adjusting $\lambda$ across the range {0, 0.1, 0.3, 0.5, 0.7, 0.9, 1.0} and $\tau$ within {0.1, 0.3, 0.5, 0.7}. Our observations predominantly support the use of the hyperparameter combination $\tau = 0.5$ and $\lambda = 0.9$ in all evaluated scenarios, as these configurations consistently achieved the highest validation accuracies.

### A.3 FURTHER EMBEDDING QUALITY ANALYSIS: SCHANE VS. CROSS-ENTROPY

In addition to the two facets of embedding quality analysis presented in the main paper, we also employ the **Isotropy Score as defined by Mu & Viswanath (2018)** to evaluate the quality of produced embeddings. The Isotropy Score measures the distribution of data in the embedding space and serves as a metric for the quality of the produced embeddings. Historically, isotropy has served as an evaluation metric for representation quality (Arora et al., 2016). This is based on the premise that widely distributed representations across different classes in the embedding space facilitate better distinction between them.

Formally, we calculate the quantitative Isotropy Score (IS) (Mu & Viswanath, 2018), which is defined as follows:

---

[2]https://github.com/microsoft/unilm/tree/master/beit3
[3]https://scikit-learn.org/stable/

$$IS(\mathcal{V}) = \frac{max_{c \subset C} \sum_{v \subset V} \exp{(C^T V)}}{min_{c \subset C} \sum_{v \subset V} \exp{(C^T V)}} \tag{5}$$

where $V$ is a set of vectors, $C$ is the set of all possible unit vectors (i.e., any $c$ so that $||c|| = 1$) in the embedding space. In practice, $C$ is approximated by the eigenvector set of $V^T V$ ($V$ are the stacked embeddings of $v$). The larger the IS value, the more isotropic an embedding space is (i.e., a perfectly isotropic space obtains an IS score of 1).

Table 4 demonstrates that the IS score for BEiT-3-SCHaNe is superior to that of BEiT-3-CE, confirming that SCHaNe produces a more isotropic semantic space. The BEiT-3-CE embeddings are more anisotropic, implying that BEiT-3-SCHaNe embeddings more distinctly separate the different classes.

