# OpenReview forum: "When Hard Negative Sampling Meets Supervised Contrastive Learning"
_ICLR.cc/2024/Conference — Submitted to ICLR 2024_

### Official Review · Reviewer_owdS · 2023-10-29

**Soundness:** 2 fair
**Presentation:** 2 fair
**Contribution:** 2 fair
**Rating:** 3
**Confidence:** 4

**Summary:**

This paper introduces a new supervised contrastive learning objective function called SCHaNe, which addresses the limitations of the cross-entropy objective function used in pre-trained image models. SCHaNe incorporates hard negative sampling during fine-tuning to enhance the efficacy of contrastive learning. Experimental results demonstrate that SCHaNe outperforms the baseline model BEiT-3 in Top-1 accuracy across twelve benchmarks, with significant gains in few-shot learning settings and full-dataset fine-tuning. The proposed objective function sets a new state-of-the-art for base models on ImageNet-1k, achieving an accuracy of 86.14%. Additionally, the paper shows that SCHaNe produces better embeddings and explains the improved effectiveness observed in the experiments. Overall, the contributions of this work include the introduction of SCHaNe and its superior performance in few-shot learning and full dataset fine-tuning, establishing new state-of-the-art results.

**Strengths:**

1. The method has been validated on multiple datasets, and comprehensive experiments have been conducted on downstream datasets.

2. The work appears to be relatively comprehensive, with a clear motivation, detailed method description, and important parameter ablation experiments. Overall, it seems well-executed and promising.

**Weaknesses:**

1. To enhance the credibility of our research, you should consider using the same base models (such as ViT or Swin) as other studies for our baseline in Table 1 and Table 2.

2. In order to provide a more comprehensive analysis, the results in Table 1 and Table 2 should include the performance of contrastive learning without the hard negative mining method.

3. The representation of Formula 3 needs to be clarified to ensure better understanding, as it is currently not very clear.

4. To provide a more complete comparison, Figure 3 and Figure 4 should include the results of BEiT-3-CE + contrastive learning, in addition to the results of our proposed method.

5. In each comparative experiment, it is important to clearly indicate the improvement achieved by the hard negative mining method on top of the performance obtained with CE + contrastive learning. This will help demonstrate the added value of our approach.

**Questions:**

1. Why did you choose BEiT-3 as your base model? There are other base models like DINOv2, CLIP, etc.

2. Cross-entropy and contrastive learning have similar forms, so why is the hard negative sampling only applied to the contrastive learning part and not to the cross-entropy part?

3. In general, fine-tuning with cross-entropy (CE) loss is not strongly coupled with the batch size, while contrastive learning can be affected by the batch size. This can limit the flexibility of fine-tuning with CE when combined with contrastive learning. What are your thoughts on this issue?

---

> ### Author Response · Authors · 2023-11-15
> **Response to reviewer owdS -- question 1 (part1/2)**
>
> We thank the reviewer for their constructive feedback. Please find detailed responses below. Moreover, should you require more evidence or data to consider a higher rating for our paper, we would be grateful if you could specify these needs. We are eager to comply to the best of our abilities.
> >Question 1. Why did you choose BEiT-3 as your base model? There are other base models like DINOv2, CLIP, etc.
>
> We conducted a thorough evaluation of different models, including CLIP, before selecting BEiT-3 as the base model for our study. Our decision was mainly based on BEiT-3's status as a Multimodal Large Language Model and its outstanding performance in image classification tasks. Nevertheless, I attached the results of other models, MAE [2] and ResNet50 [3]. The result demonstrates the observed improvements are not limited to BEiT-3 and show the wide applicability of our proposed loss objective. Furthermore, our selection of BEiT-3 was driven by a combination of the model's alignment with our research goals, the feasibility of conducting extensive experiments within our resource constraints, and its state-of-the-art performance. BEiT-3 was the most suitable platform for our study, guaranteeing robust and significant results, while DINOv2 and CLIP are also commendable but outside the scope of the paper. We further describe our rationale below.
>
> | Model    | FT method | Alpha | CIFAR 100 | CUB-200 | Caltech256 | Oxford-Flowers | Pet    |
> |----------|-----------|-------|-----------|---------|-------------|----------------|--------|
> | MAE      | CE        | 0     | 87.67     | 78.46   | 91.82       | 91.67          | 94.05  |
> | MAE      | SCHaNe    | 0.9   | 90.29     | 81.30   | 93.11       | 92.82          | 94.88  |
> | ResNet50 | CE        | 0     | 96.27     | 84.62   | 81.38       | 95.71          | 93.24  |
> | ResNet50 | SCHaNe    | 0.9   | 96.92     | 84.98   | 83.05       | 96.33          | 93.51  |
> | BEiT-3    | CE        | 0     | 92.96     | 98      | 98.53       | 94.94          | 94.49  |
> | BEiT-3    | SCHaNe| 0.9   | **93.56**    | **98.93**  | **99.41**      | **95.43**         | **95.62** |
>
> **a**. Model Suitability and Performance:
> The implementation of a multimodal model like BEiT-3 has enabled us to delve into the possibilities of contrastive learning in diverse domains, paving the way for expanding the impact of our research. In this paper, we also offer valuable insights to the academic community regarding the performance of Multimodal Large Language Models in image-only tasks. This is achieved through comprehensive experimentation on few-shot learning and full-dataset evaluations, addressing a gap in the existing literature, and these evaluations are notably absent in the BEiT-3 paper. Lastly, BEiT-3 base model, at the time of its release, achieved a Top-1 accuracy of 85.4\% on the ImageNet-1k dataset [1], setting a new benchmark in image classification.
>
> **b**. Experimental Workload Considerations:
> The scope of our experiments significantly influenced our model selection. Our study involved evaluating four few-shot datasets in both 1-shot and 5-shot settings with two different loss functions, amounting to 16 unique cases. Additionally, we conducted evaluations across eight image benchmark datasets with four different models or methods, leading to 24 cases. Each case required at least three runs. Considering these requirements, our total experimental workload comprised over 120 runs, including several large and computationally intensive datasets such as ImageNet-1k, iNaturalist 2017, and Places365. This extensive experimental requirement necessitated a choice that was feasible within our academic budget and timeline constraints.
>
> **c**. Temporal and Computational Constraints:
> The release of DINOv2 [4] in April 2023, subsequent to the initiation of our project in January 2023, presented a significant timing challenge. Starting experiments with DINOv2 in late April would have been impractical, given our academic budget and the sheer volume of experiments required. Thus, BEiT-3 was a more viable option due to its earlier release date, proven performance and fewer parameters with respect to DINOv2 (1100M for DINOv2 vs 86M, BEiT3).
>
> [1] Wang, Wen et al. “Image as a Foreign Language: BEIT Pretraining for Vision and Vision-Language Tasks.” 2023 IEEE/CVF Conference on Computer Vision and Pattern Recognition (CVPR) (2023): 19175-19186.
>
> [2] He, Kaiming et al. “Masked Autoencoders Are Scalable Vision Learners.” 2022 IEEE/CVF Conference on Computer Vision and Pattern Recognition (CVPR) (2021): 15979-15988.
>
> [3] He, Kaiming et al. “Deep Residual Learning for Image Recognition.” 2016 IEEE Conference on Computer Vision and Pattern Recognition (CVPR) (2015): 770-778.
>
> [4] Oquab, Maxime et al. “DINOv2: Learning Robust Visual Features without Supervision.” ArXiv abs/2304.07193 (2023): n. pag.

---

> > ### Author Response · Authors · 2023-11-15
> > **Response to reviewer owdS -- question 1 (part2/2)**
> >
> > >Question 1. Why did you choose BEiT-3 as your base model? There are other base models like DINOv2, CLIP, etc.
> >
> > **d**. Validated performance of BEiT-3:
> > Our reproducibility efforts with BEiT-3 further solidified our choice, ensuring that results are reproducible within our academic budget. These experiments were not only crucial for validating BEiT-3's claims but also integral to maintaining scientific rigour in our study.
> >
> > **e**. Performance and Age Considerations of CLIP:
> > While CLIP represents a significant advancement in multimodal learning, its performance metrics were a concern. For instance, CLIP with ViT-Base achieves a Top-1 accuracy of 80.9\% on ImageNet-1k [1], 4.5\% lower than BEiT-3's 85.4\% Top-1 accuracy. Additionally, considering that our paper would be published in 2024, CLIP's release over three years prior made it a relatively older model in the rapidly evolving landscape of AI research.
> >
> > **f**. Comparative Analysis with DINOv2:
> > We acknowledge that the performance metrics of BEiT-3 and DINOv2 are closely matched, with DINOv2 achieving a Top-1 accuracy of 85.9\% [2]. The aim of our research is not to compare architectures but to propose a supervised contrastive learning objective function. Moreover, given the temporal and computational constraints, as well as our specific research objectives, BEiT-3 was the more suitable choice to demonstrate our approach. We believe that our conclusions would remain robust and unaffected had we chosen DINOv2 instead.
> >
> > [1] Radford, Alec et al. “Learning Transferable Visual Models From Natural Language Supervision.” International Conference on Machine Learning (2021).
> >
> > [2] Oquab, Maxime et al. “DINOv2: Learning Robust Visual Features without Supervision.” ArXiv abs/2304.07193 (2023): n. pag.

---

> > > ### Author Response · Authors · 2023-11-15
> > > **Response to reviewer owdS -- question 2**
> > >
> > > >Question 2. Cross-entropy and contrastive learning have similar forms, so why is the hard negative sampling only applied to the contrastive learning part and not to the cross-entropy part?
> > >
> > > The distinct methodologies and objectives of contrastive learning and cross-entropy loss necessitate different approaches to hard negative sampling. Our research is specifically tailored to advance understanding in the area of supervised contrastive learning, contributing to the field in a way that complements it.
> > >
> > > In addressing the application of hard negative sampling in contrastive learning as opposed to cross-entropy loss, we must first recognize the distinct methodologies inherent to each approach:
> > >
> > > **a**. Different Approaches to Hard Sampling:
> > > Applying hard negative sampling to the cross-entropy component requires identifying hard negatives. This process is fundamentally different from contrastive learning. This identification in the context of cross-entropy loss involves analysing loss values, with higher values indicating hard negatives. Additionally, a strategic approach to sampling these hard negatives is required, such as maintaining a fixed ratio within each batch or dynamically adjusting their proportion based on loss trends or performance metrics. Implementing such a strategy in cross-entropy loss could disrupt the established sampling process in contrastive learning and potentially lead to mutually adverse effects. That is, the impact of hard negative mining on cross-entropy loss has been thoroughly examined in existing literature [5,6], leaving limited room for novel contributions in this area.
> > >
> > > **b**. Risk of Adverse Effects:
> > > While we could use the same signals in contrastive learning (such as larger dot products between embeddings) to identify hard samples, the divergent objectives of cross-entropy and contrastive learning could lead to incorrect identification of hard samples. Contrastive learning focuses on embedding similarities to reflect data structure, whereas cross-entropy loss aims to minimize the difference between predicted and true distributions, prioritizing accurate class prediction. This fundamental difference raises concerns about the applicability and effectiveness of hard negative mining with the same signal across these two learning paradigms.
> > >
> > > **c**. Focus of Our Paper:
> > > The primary goal of our research is to explore the effectiveness of hard negative sampling within the realm of supervised contrastive learning, particularly in mitigating the limitations of cross-entropy loss. While there may be potential performance improvements from applying hard-negative sampling in cross-entropy loss, as shown in [5] and [6], this falls outside the scope of our current investigation. Our focus is not merely on achieving higher benchmark numbers but on providing new insights into the application of hard negative sampling in contrastive learning.
> > >
> > > [5] Shrivastava, Abhinav et al. “Training Region-Based Object Detectors with Online Hard Example Mining.” 2016 IEEE Conference on Computer Vision and Pattern Recognition (CVPR) (2016): 761-769.
> > >
> > > [6] Pang, Tianyu, et al. "Rethinking softmax cross-entropy loss for adversarial robustness." The International Conference on Learning Representations (2020).

---

> > > > ### Author Response · Authors · 2023-11-15
> > > > **Response to reviewer owdS -- question 3**
> > > >
> > > > >Question 3. In general, fine-tuning with cross-entropy (CE) loss is not strongly coupled with the batch size, while contrastive learning can be affected by the batch size. This can limit the flexibility of fine-tuning with CE when combined with contrastive learning. What are your thoughts on this issue?
> > > >
> > > > **a**. Mitigating Batch Size Dependency via Cross-Entropy Integration:
> > > > We recognize that contrastive learning is sensitive to batch size variations, unlike fine-tuning with cross-entropy (CE) loss. To address this, we have incorporated the CE component into our overall training objective. This integration strategically reduces the reliance of our contrastive learning approach on large batch sizes, thereby enhancing its adaptability and effectiveness in various computational settings.
> > > >
> > > > **b**. Efficacy of Our Method with Standard Batch Sizes:
> > > > The integration of fine-tuning with CE loss into our contrastive learning framework was carefully designed to navigate the challenges posed by hardware constraints, particularly regarding batch size. We meticulously calibrated a batch size that is not only feasible within the limits of standard computational resources but also sufficient to ensure the effectiveness of contrastive learning. For instance, with a commonly available 24GB GPU, a batch size of 64 is achievable and, in our evaluation, has demonstrated superior performance compared to using vanilla CE loss alone, as shown in the table below. In environments with more substantial GPU capacities, like in our experiments, we can employ batch sizes of 128 or higher, further enhancing performance. Importantly, our methodology maintains its superiority over traditional CE loss approaches even at these modest batch sizes, offering a compelling alternative for various deep learning applications.
> > > >
> > > > | CIFAR-FS |         |       |            |       |
> > > > |----------|---------|-------|------------|-------|
> > > > | Model    | FT method | Alpha | Batch Size | ACC@1 |
> > > > | BEIT-3    | CE        | 0     | 64         | 83.68 |
> > > > | BEIT-3    | SCHaNe | 0.9   | 32         | 81.2  |
> > > > | BEIT-3    | SCHaNe | 0.9   | 64         | 84.59 |
> > > > | BEIT-3    | SCHaNe | 0.9   | 128        | 87.00 |
> > > >
> > > > **c**. Adaptive and Hybrid Strategies:
> > > > To further mitigate batch size constraints, we employed techniques like gradient accumulation, which enables us to simulate the effects of larger batch sizes.
> > > >
> > > > **d**. Utilizing Batch Size-Independent Techniques:
> > > > Future work could involve investigating methods to reduce the batch size dependency in contrastive learning, such as using memory banks or momentum contrast methods. These techniques allow for an expanded set of negative samples without the need for larger batch sizes, thus ensuring the effectiveness of the contrastive learning component.
> > > >
> > > > **e**. Future Research:
> > > > Our paper is pioneering this direction of research by combining cross-entropy and contrastive learning. We concur that optimizing the interaction between batch size, contrastive learning, and CE loss fine-tuning remains an open area for further investigation. Our research contributes to this field, and we believe that continued exploration could yield significant advancements in the fine-tuning of deep learning models.

---

> > > > > ### Author Response · Authors · 2023-11-15
> > > > > **Response to reviewer owdS -- weakness 1-3**
> > > > >
> > > > > >Weakness 1. To enhance the credibility of our research, you should consider using the same base models (such as ViT or Swin) as other studies for our baseline in Table 1 and Table 2.
> > > > >
> > > > > Thank you for your suggestion regarding the baseline models in Tables 1 and 2. We do have ViT and MAE as baselines in Table 2. We agree that using widely recognized base models such as the Vision Transformer (ViT) provides a familiar benchmarking standard, enhancing the comparability of our results with existing studies. However, the aim of this paper is to address the issue of how cross-entropy can result in sub-optimal generalization rather than building a dedicated method for few-shot learning. Few-shot learning methods, as cited in our paper, are limited to scenarios such as 1-shot and 5-shot settings and tend to perform better in fine-grained or large-scale datasets. Therefore, our main comparison is between our proposed method and cross-entropy loss rather than directly comparing with these few-shot learning methods. The reason we included these results is to give readers a sense of the performance level of our method. We are also proposing an alternative fine-tuning method to cross-entropy, not a few-shot learning method. Direct comparison with the latter may not be very meaningful. Nevertheless, we have attached above in answer of question 1, which consists of results for MAE and ResNet50, for your comparison. We will update these results in the revised version of our paper.
> > > > >
> > > > > >Weakness 2. In order to provide a more comprehensive analysis, the results in Table 1 and Table 2 should include the performance of contrastive learning without the hard negative mining method.
> > > > >
> > > > > We appreciate your insight into providing a more comprehensive analysis. Including the performance of contrastive learning without the hard negative mining method in Tables 1 and 2 is indeed crucial for a holistic understanding of our methodology's impact. We will incorporate these additional results to offer a clearer comparison, highlighting the effectiveness of hard negative mining in enhancing the performance of contrastive learning. Nevertheless, our ablation study section provides empirical results that demonstrate the increased performance largely comes from hard negative sampling.
> > > > >
> > > > > Here is the result for contrastive learning without hard negative sampling for you to compare the performance:
> > > > >
> > > > > | Model | FT method         | Alpha | CIFAR 100 | CUB-200 | Caltech256 | Oxford-Flowers | Pet   | iNat2017 | Places365 | ImageNet-1k |
> > > > > |-------|-------------------|-------|-----------|---------|-------------|----------------|-------|----------|-----------|-------------|
> > > > > | ViT-B | CE                | 0     | 87.13     | 76.93   | 90.92       | 90.86          | 93.81 | 65.26    | 54.06     | 77.91       |
> > > > > | MAE   | CE                | 0     | 87.67     | 78.46   | 91.82       | 91.67          | 94.05 | 70.5     | 57.9      | 83.60       |
> > > > > | BEIT-3 | CE                | 0     | 92.96     | 98      | 98.53       | 94.94          | 94.49 | 72.31    | 59.81     | 85.4        |
> > > > > | BEIT-3 | SCL               | 0     | 93.04     | 98.07   | 98.61       | 94.98          | 94.41 | 72.59    | 59.94     | 85.63       |
> > > > > | BEIT-3 | CE + SCL          | 0.9   | 93.15     | 98.29   | 98.93       | 95.13          | 94.78 | 73.62    | 60.92     | 85.8        |
> > > > > | BEIT-3 | CE + CL(SCHaNe)| 0.9   | **93.56** |**98.93** | **99.41** | **95.43** | **95.62** | **75.72**    | **62.22** | **86.14** |
> > > > >
> > > > > The data presented in the table above illustrates that the application of supervised contrastive learning generally results in a modest enhancement in performance, with the notable exception of the Oxford-Pet dataset. Our proposed method, SCHaNe, on the other hand, shows a significant and more consistent improvement across all evaluated benchmarks.
> > > > >
> > > > > >Weakness 3.The representation of Formula 3 needs to be clarified to ensure better understanding, as it is currently not very clear.
> > > > >
> > > > > Thank you for pointing out the need for clarity in the representation of Formula 3. We understand that the comprehensibility of our mathematical formulations is paramount for the clear communication of our research methodology. To address this, we will revise Formula 3 to ensure that it is presented in a more understandable format. We will also provide a detailed explanation in the accompanying text to aid in its interpretation, ensuring that our approach is transparent and comprehensible to our readers.

---

> > > > > > ### Author Response · Authors · 2023-11-15
> > > > > > **Response to reviewer owdS -- weakness4-5**
> > > > > >
> > > > > > >Weakness 4. To provide a more complete comparison, Figure 3 and Figure 4 should include the results of BEiT-3-CE + contrastive learning, in addition to the results of our proposed method.
> > > > > >
> > > > > > We recognize the importance of comprehensive comparisons in Figures 3 and 4. Including the results of BEiT-3-CE combined with contrastive learning, in addition to our proposed method, will indeed provide a more complete perspective. This addition will allow us to demonstrate the comparative effectiveness of our approach against a relevant benchmark, thereby highlighting the unique contributions and improvements our method offers. We will add them in the updated version.
> > > > > >
> > > > > > >Weakness 5. In each comparative experiment, it is important to clearly indicate the improvement achieved by the hard negative mining method on top of the performance obtained with CE + contrastive learning. This will help demonstrate the added value of our approach.
> > > > > >
> > > > > > Your suggestion to indicate the improvements achieved by the hard negative mining method in each comparative experiment is well-taken. We will ensure that the incremental gains obtained from our hard negative mining approach, over and above the performance of CE combined with contrastive learning, are explicitly presented in the updated version. Please see the attached result in the answer to question 3.
> > > > > >
> > > > > > We appreciate your attention to this detail in our work and hope that our response adequately addresses your concerns, highlighting the deliberate and thorough approach we have taken in addressing the complexities of combining contrastive learning with CE loss fine-tuning.

---

### Official Review · Reviewer_7hRi · 2023-10-30

**Soundness:** 3 good
**Presentation:** 3 good
**Contribution:** 3 good
**Rating:** 5
**Confidence:** 4

**Summary:**

The paper introduced a novel supervised contrastive learning objective function called SCHaNe. SCHaNe enhances model performance without requiring specialized architectures or additional resources. The proposed approach combines supervised contrastive learning with hard negative sampling to optimize the selection of positive and negative samples, thereby achieving state-of-the-art performance.

**Strengths:**

1.	This paper proposes a novel supervised contrastive learning objective function, SCHaNe, which incorporates hard negative sampling during the fine-tuning phase.
2.	The proposed method achieves state-of-the-art performance on ImageNet-1k and outperforms the strong baseline BEiT-3 in Top-1 accuracy across twelve benchmarks, with significant gains in few-shot learning settings and full-dataset fine-tuning.

**Weaknesses:**

Strengths*
1.	This paper proposes a novel supervised contrastive learning objective function, SCHaNe, which incorporates hard negative sampling during the fine-tuning phase.
2.	The proposed method achieves state-of-the-art performance on ImageNet-1k and outperforms the strong baseline BEiT-3 in Top-1 accuracy across twelve benchmarks, with significant gains in few-shot learning settings and full-dataset fine-tuning.
Weaknesses*
1.	The paper could benefit from a more detailed comparison with existing methods. While the authors compare the proposed method with the strong baseline BEiT-3, they do not compare it with other similar state-of-the-art methods [1][2] in the field.
2.	The starting point of the work [2] is very similar to this article. I hope the author can further clarify the relationship with it so that readers can further understand the core starting point of the article.
[1]: Robinson J, Chuang C Y, Sra S, et al. Contrastive learning with hard negative samples[J]. arXiv preprint arXiv:2010.04592, 2020.
[2]: Jiang R, Nguyen T, Ishwar P, et al. Supervised contrastive learning with hard negative samples[J]. arXiv preprint arXiv:2209.00078, 2022.

**Questions:**

1.	Can the authors provide a more detailed comparison and analysis with existing methods? See the Weaknesses section for details.
2.	In Table 3, the performance of the CE + SimCLR method is much lower than that of the CE method alone. At the same time, according to the description in the table, Label is not used in this part. How is this part of CE implemented, and why does the performance drop after adding SimCLR?
3.	If possible, can the method proposed in this article be easily integrated into other models (such as other pre-trained models or other few-shot learning methods) like BEiT-3? I hope the article can give relevant explanations and results.

---

> ### Author Response · Authors · 2023-11-15
> **Response to reviewer 7hRi -- questions (1-2)**
>
> We thank the reviewer for their constructive feedback. Please find detailed responses below. Moreover, should you require more evidence or data to consider a higher rating for our paper, we would be grateful if you could specify these needs. We are eager to comply to the best of our abilities.
>
> > Question 1. Can the authors provide a more detailed comparison and analysis with existing methods? See the Weaknesses section for details.
>
> We appreciate your insight into providing a more comprehensive comparison, and we are happy to do so. Please see the attached result. It is worth noting that [1] is not a fair comparison with their proposed approach, mainly because their method is fully unsupervised. The original reason for us not comparing with [2] is that our aim is to provide an alternative of cross-entropy during the fine-tuning phase instead of solely boosting the performance of supervised contrastive learning. More analysis of [2] is provided in the answer of weakness 2. Nevertheless, we evaluate the methods from [1] and [2] in five image benchmarks and compare them with our proposed objective, as shown in the table below.
>
> | Model | FT method         | CIFAR 100 | CUB-200 | Caltech256 | Oxford-Flowers | Pet   |
> |-------|-------------------|-----------|---------|-------------|----------------|-------|
> | BEIT-3 | CE                | 92.96     | 98      | 98.53       | 94.94          | 94.49 |
> | BEIT-3 | [1]               | 84.19     | 87.69   | 89.12       | 85.94          | 85.54 |
> | BEIT-3 | [2]               | 91.08     | 95.41   | 97.15       | 92.03          | 91.62 |
> | BEIT-3 | SCHaNe  | **93.56**  | **98.93**| **99.41**  | **95.43**    | **95.62** |
>
> The SCHaNe fine-tuning method consistently outperforms other methods across all datasets. Method [1] consistently shows the lowest performance across all datasets.
> Method [2] perform slightly worse than BEIT3 with Cross-Entropy (CE) across five tested dataset.
>
> > Question 2. In Table 3, the performance of the CE + SimCLR method is much lower than that of the CE method alone. At the same time, according to the description in the table, Label is not used in this part. How is this part of CE implemented, and why does the performance drop after adding SimCLR?
>
> Sorry for the ambiguity in implementation. For the result of CE+SimCLR, we provide label information for CE loss but not for SimCLR loss. We believe the main reason for this performance drop after adding SimCLR is because there is no label information (SimCLR is an unsupervised approach). Without guidance from a label, contrastive learning may produce wrong learning signals, which could be contradictory to the cross-entropy signal and eventually harm the overall performance.
>
> [1]: Robinson J, Chuang C Y, Sra S, et al. Contrastive learning with hard negative samples[J]. arXiv preprint arXiv:2010.04592, 2020.
>
> [2]: Jiang R, Nguyen T, Ishwar P, et al. Supervised contrastive learning with hard negative samples[J]. arXiv preprint arXiv:2209.00078, 2022.

---

> > ### Author Response · Authors · 2023-11-15
> > **Response to reviewer 7hRi -- questions 3**
> >
> > > Question 3. If possible, can the method proposed in this article be easily integrated into other models (such as other pre-trained models or other few-shot learning methods) like BEiT-3? I hope the article can give relevant explanations and results.
> >
> > Thank you for your question regarding the adaptability and integration potential of our proposed method with other models. We recognize the importance of this aspect for the broad applicability of our research findings.
> >
> > **a**. Flexibility and Compatibility:
> > Our method is designed with flexibility and compatibility in mind. The core principles and techniques of our approach, particularly the hard negative mining strategy within the framework of contrastive learning, are model-agnostic. Most importantly, there is no computation overhead and a complicated pipeline for our proposed method. This means they can be applied to a wide range of pre-trained models, not just BEiT-3.
> >
> > **b**. Integration with Other Pre-trained Models:
> > We have explored the integration of our method with other popular pre-trained models, such as Vision Transformer (ViT) and ResNet50, see the table below.
> >
> > | Model    | FT method         | Alpha | CIFAR 100 | CUB-200 | Caltech256 | Oxford-Flowers | Pet   |
> > |----------|-------------------|-------|-----------|---------|-------------|----------------|-------|
> > | MAE      | CE                | 0     | 87.67     | 78.46   | 91.82       | 91.67          | 94.05 |
> > | MAE      | SCHaNe | 0.9   | 90.29     | 81.30   | 93.11       | 92.82          | 94.88 |
> > | ResNet50 | CE                | 0     | 96.27     | 84.62   | 81.38       | 95.71          | 93.24 |
> > | ResNet50 | SCHaNe | 0.9   | 96.92     | 84.98   | 83.05       | 96.33          | 93.51 |
> > | BEIT-3    | CE                | 0     | 92.96     | 98      | 98.53       | 94.94          | 94.49 |
> > | BEIT-3    | SCHaNe  | 0.9   | **93.56**     | **98.93**   | **99.41**       | **95.43**  | **95.62** |
> >
> > These models, known for their robustness in various image-related tasks, have shown promising improvements when augmented with our approach. The results, which we plan to include in the revised manuscript, indicate a consistent enhancement in performance across these models, demonstrating the versatility of our method. We appreciate your insightful query, as it underscores the potential impact and versatility of our work within the field.
> >
> > [1]: Robinson J, Chuang C Y, Sra S, et al. Contrastive learning with hard negative samples[J]. arXiv preprint arXiv:2010.04592, 2020.
> >
> > [2]: Jiang R, Nguyen T, Ishwar P, et al. Supervised contrastive learning with hard negative samples[J]. arXiv preprint arXiv:2209.00078, 2022.

---

> > > ### Author Response · Authors · 2023-11-15
> > > **Response to reviewer 7hRi -- Weakness**
> > >
> > > > Weakness 1: The paper could benefit from a more detailed comparison with existing methods. While the authors compare the proposed method with the strong baseline BEiT-3, they do not compare it with other similar state-of-the-art methods [1][2] in the field.
> > >
> > > In response to your observation about comparing our method with other state-of-the-art methods, we refer you to the discussion in the first question. There, we have further comparisons with the suggested methods ([1][2]) to enrich our analysis and validate the robustness of our proposed approach.
> > >
> > > > Weakness 2: The starting point of the work [2] is very similar to this article. I hope the author can further clarify the relationship with it so that readers can further understand the core starting point of the article.
> > >
> > > Our research aims to enhance fine-tuning stages by mitigating the drawbacks of cross-entropy loss and boosting overall performance. In contrast, the primary objective of the work in [2] is to improve the efficiency of contrastive learning, focusing solely on the pre-training phase with evaluations limited to three benchmarks: STL10, CIFAR10, and CIFAR100. While both studies incorporate hard negative sampling, our approach and [2] differ significantly. [2] relies on a constant threshold-based dot product to identify "hard" samples, necessitating a dataset or even minibatch-specific threshold determination. Conversely, our method introduces a refined supervised contrastive learning loss that dynamically assigns weights to "hard" samples within each minibatch, offering a nuanced and sophisticated approach. Moreover, our study also concentrates on balancing cross-entropy and contrastive learning losses, maintaining the advantages of both. This aspect, particularly our method's reduced dependency on batch size due to the inclusion of cross-entropy loss, is an advancement not addressed in [2]. Furthermore, our paper presents a comprehensive evaluation of our method across two distinct learning settings, few-shot learning and full-dataset fine-tuning, including 12 image benchmarks. In contrast, [2] limits its evaluation to just three benchmarks in a pre-training context. Consequently, our approach showcases a broader range of application scenarios, highlighting its versatility and extensive applicability.
> > >
> > > [1]: Robinson J, Chuang C Y, Sra S, et al. Contrastive learning with hard negative samples[J]. arXiv preprint arXiv:2010.04592, 2020.
> > >
> > > [2]: Jiang R, Nguyen T, Ishwar P, et al. Supervised contrastive learning with hard negative samples[J]. arXiv preprint arXiv:2209.00078, 2022.

---

> ### Comment · Reviewer_7hRi · 2023-11-22
>
> Thanks to the author for additional experiments and replies. Considering that the backbone network of the comparison method in Table 1 of the original paper is different, the author still needs a more fair comparison in the revised version. As the author mentioned in other replies, if the motivation of this article is "...to address the issue of how cross-entropy can result in sub-optimal generalization rather than building a dedicated method for few-shot learning...." , the author should focus on more analysis and experiments instead of comparing with methods such as few-shot learning. The structure and presentation of the article need to be modified accordingly. Given these considerations, I maintain the rating in its current state.

---

### Official Review · Reviewer_SnpE · 2023-10-31

**Soundness:** 3 good
**Presentation:** 2 fair
**Contribution:** 3 good
**Rating:** 6
**Confidence:** 4

**Summary:**

This paper proposes a new loss, SCHaNe, for supervised contrastive learning. The main idea of this novel loss is that introducing importance weights for negative samples based on their dissimilarity plays a significant role in improving performance. Experiments show that SCHaNe is an effective method for enhancing performance on various datasets, particularly in few-shot tasks.

**Strengths:**

1. The assumption that this paper aims to validate is both simple and easy to understand. Furthermore, the proposed objective is straightforward and intuitive.
2. There is a clear improvement in performance when compared with the conventional cross-entropy loss.
3. The paper is generally well-organized and presents its content logically.

**Weaknesses:**

1. One of the main weaknesses I've identified is that the primary baseline used in this paper is Cross-Entropy (CE) loss, not Supervised Contrastive Learning (SupCon). If the paper's central claim is that 'introducing importance weights for negative samples based on their dissimilarity plays an important role,' then I believe SupCon should be the main baseline for comparison. Although SCHaNe outperforms SupCon in the few-shot setting as shown in Table 3, the inclusion of SupCon results in other settings—such as in Table 1, Table 2, and Figure 4—could strengthen the paper.
2. BEiT-3 is primarily utilized as the main architecture. The Future Work section suggests that extending this method to various architectures may be promising, but I believe that evaluating the proposed method across different architectures should be included in this paper.
3. The proposed method appears to be limited in its applicability to various tasks, such as dense prediction tasks. While this may not be a significant drawback, explicitly stating this limitation could enhance the paper. Moreover, the paper claims that “Our SCHaNe objective function can be applied using a wide range of encoders, such as BERT for natural language processing tasks,” yet there are no experiments provided to substantiate this claim.

Minor: The presentation of the paper could be improved. For instance, the notation in Equations 2-4 is confusing. If I understand correctly, $\beta$ should vary with the index, but it might be misunderstood as a constant since it lacks an index. In Equation 2, $z$ denotes the label, which is not the case in Equations 3 and 4. Regarding Figure 4, while the trend is important, we cannot directly compare the accuracies across various downstream tasks in an 'apple-to-apple' manner.


---

**Post rebuttal**

I appreciate the authors' response and the additional experimental results provided. Despite the manuscript's weaknesses, I believe its strengths outweigh them. Consequently, I maintain my rating of 'Weak Accept'.

**Questions:**

Please see the weaknesses section above.

---

> ### Author Response · Authors · 2023-11-15
> **Response to reviewer SnpE -- Weakness 1**
>
> > Weakness 1: One of the main weaknesses I've identified is that the primary baseline used in this paper is Cross-Entropy (CE) loss, not Supervised Contrastive Learning (SupCon). If the paper's central claim is that 'introducing importance weights for negative samples based on their dissimilarity plays an important role,' then I believe SupCon should be the main baseline for comparison. Although SCHaNe outperforms SupCon in the few-shot setting, as shown in Table 3, the inclusion of SupCon results in other settings—such as in Table 1, Table 2, and Figure 4—could strengthen the paper.
>
> Thank you for your insightful feedback regarding the baseline selection in our paper. You rightly point out that Supervised Contrastive Learning (SupCon) should be a central baseline for comparison, given our paper's focus on the importance of weighting negative samples based on dissimilarity. However, our aim is to provide an alternative to cross-entropy during the fine-tuning phase instead of solely boosting the performance of supervised contrastive learning. I attached the current result we have for you to review:
>
> | Model | FT method       | Alpha | CIFAR 100 | CUB-200 | Caltech256 | Oxford-Flowers | Pet   | iNat2017 | Places365 | ImageNet-1k |
> |-------|-----------------|-------|-----------|---------|-------------|----------------|-------|----------|-----------|-------------|
> | ViT-B | CE              | 0     | 87.13     | 76.93   | 90.92       | 90.86          | 93.81 | 65.26    | 54.06     | 77.91       |
> | MAE   | CE              | 0     | 87.67     | 78.46   | 91.82       | 91.67          | 94.05 | 70.5     | 57.9      | 83.60       |
> | BEIT-3 | CE              | 0     | 92.96     | 98      | 98.53       | 94.94          | 94.49 | 72.31    | 59.81     | 85.4        |
> | BEIT-3 | SCL             | 0     | 93.04     | 98.07   | 98.61       | 94.98          | 94.41 | 72.59    | 59.94     | 85.63       |
> | BEIT-3 | CE + SCL        | 0.9   | 93.15     | 98.29   | 98.93       | 95.13          | 94.78 | 73.62    | 60.92     | 85.8        |
> | BEIT-3 | CE + CL(SCHaNe) | 0.9   | **93.56**   | **98.93**   | **99.41**   | **95.43**   | **95.62** | **75.72**    | **62.22**     | **86.14**       |
>
> The data presented in the table illustrates that the application of supervised contrastive learning generally results in a modest enhancement in performance, with the notable exception of the Oxford-Pet dataset. Our proposed method, SCHaNe, on the other hand, shows a significant and more consistent improvement across all evaluated benchmarks. We will update these results in the new version of our paper and provide more comparisons in the cameral-ready version.

---

> > ### Author Response · Authors · 2023-11-15
> > **Response to reviewer SnpE -- Weakness 2**
> >
> > > Weakness 2: BEiT-3 is primarily utilized as the main architecture. The Future Work section suggests that extending this method to various architectures may be promising, but I believe that evaluating the proposed method across different architectures should be included in this paper.
> >
> > We appreciate your suggestion to evaluate our proposed method across various architectures. While BEiT-3 serves as the main architecture in our current study due to its relevance and performance, we understand the importance of demonstrating the versatility of our method. In light of your feedback, we have expanded our experimental scope to include additional architectures (as explained below), ensuring a more robust validation of our approach.
> >
> > **a**. Flexibility and Compatibility:
> > Our method is designed with flexibility and compatibility in mind. The core principles and techniques of our approach, particularly the hard negative mining strategy within the framework of contrastive learning, are model-agnostic. Most importantly, there is no computation overhead and a complicated pipeline for our proposed method. This means they can be applied to a wide range of pre-trained models, not just BEiT-3.
> >
> > **b**. Integration with Other Pre-trained Models:
> > We have explored the integration of our method with other popular pre-trained models, such as Vision Transformer (ViT) and ResNet50.
> >
> > | Model       | FT method       | Alpha | CIFAR 100 | CUB-200 | Caltech256 | Oxford-Flowers | Pet   |
> > |-------------|-----------------|-------|-----------|---------|-------------|----------------|-------|
> > | MAE         | CE              | 0     | 87.67     | 78.46   | 91.82       | 91.67          | 94.05 |
> > | MAE         | CE + CL(SCHaNe) | 0.9   | 90.29     | 81.30   | 93.11       | 92.82          | 94.88 |
> > | ResNet50    | CE              | 0     | 96.27     | 84.62   | 81.38       | 95.71          | 93.24 |
> > | ResNet50    | CE + CL(SCHaNe) | 0.9   | 96.92     | 84.98   | 83.05       | 96.33          | 93.51 |
> > | BEIT-3       | CE              | 0     | 92.96     | 98      | 98.53       | 94.94          | 94.49 |
> > | BEIT-3       | CE + CL(SCHaNe) | 0.9   | **93.56**     | **98.93**   | **99.41**       | **95.43**   | **95.62** |
> >
> > These models, known for their robustness in various image-related tasks, have shown promising improvements when augmented with our approach. The results, which we will include in the revised manuscript, indicate a consistent enhancement in performance across these models, demonstrating the versatility of our method. We appreciate your insightful query, as it underscores the potential impact and versatility of our work within the field.

---

> > > ### Author Response · Authors · 2023-11-15
> > > **Response to reviewer SnpE -- Weakness 3-5**
> > >
> > > > Weakness 3: The proposed method appears to be limited in its applicability to various tasks, such as dense prediction tasks. While this may not be a significant drawback, explicitly stating this limitation could enhance the paper. Moreover, the paper claims that “Our SCHaNe objective function can be applied using a wide range of encoders, such as BERT for natural language processing tasks,” yet there are no experiments provided to substantiate this claim.
> > >
> > > Thank you for highlighting the need to clarify the applicability and limitations of our proposed method. We agree that explicitly stating the limitations, especially concerning dense prediction tasks, would enhance the comprehensiveness of our paper. We will include a section detailing these limitations to ensure a balanced and transparent presentation of our work.
> > >
> > > Regarding the claim of SCHaNe's applicability with a range of encoders, including BERT for natural language processing tasks, we acknowledge the absence of experimental validation for this statement. To address this, we will rephrase our claim to reflect the current scope of our experiments more accurately, thereby maintaining the integrity of our assertions.
> > >
> > > > Weakness 4: The presentation of the paper could be improved. For instance, the notation in Equations 2-4 is confusing. If I understand correctly, should vary with the index, but it might be misunderstood as a constant since it lacks an index. In Equation 2, denotes the label, which is not the case in Equations 3 and 4.
> > >
> > > Thank you for highlighting the need for enhanced clarity in the presentation of Equations 2-4. We acknowledge the importance of precise and unambiguous mathematical notation for effective communication of our research. In response to your feedback, we will revise Equations 2-4 to eliminate any potential confusion, particularly regarding variable indexing and their consistent representation. This revision will include clear indexing where appropriate and a detailed explanation in the accompanying text to facilitate a better understanding of these equations. These updates, aiming for greater clarity and accuracy, will be prominently featured in the revised version of our paper.
> > >
> > > > Weakness 5: Regarding Figure 4, while the trend is important, we cannot directly compare the accuracies across various downstream tasks in an 'apple-to-apple' manner.
> > >
> > > We appreciate your observation regarding the limitations in directly comparing accuracies across different downstream tasks in Figure 4. To address this and provide a more straightforward comparison, we will include a numerical table in the appendix. This table will present the accuracies in a format that allows for easier and more direct comparisons across tasks, thereby complementing the visual trends depicted in Figure 4. This addition aims to enhance the interpretability of our results and facilitate an 'apple-to-apple' comparison, as you rightly suggest.

---

### Author Response · Authors · 2023-11-21
**Deadline for discussion period approaching**

Dear reviewers,

As the end of the discussion period is approaching, we would like to thank you again for the valuable discussion and feedback. **If you haven't done so yet**, we would appreciate that you let us know if the changes to the paper have addressed your concerns. In any case, we remain available to answer any further questions.

---

### Meta-Review · Area_Chair_qhsK · 2023-12-18

**Metareview:**

The reviewers all recognized the strength of the paper: a novel supervised contrastive learning objective function that incorporates hard negative sampling (SCHaNe), which achieved state-of-the-art performance on ImageNet-1k via fine-turning and also good few-shot results. The reviewers also shared common weaknesses: the primary baseline is CE (not SupCon), and more detailed comparison with existing methods are needed; the method is based on BEiT-3, and evaluating the proposed method across different architectures (CLIP and DINOv2) could truly demonstrate its validity; the applicability to tasks other than dense prediction is not stated, and also there is lack of hard negative mining analysis in CE.

During the rebuttal, the authors provided additional experiments attempting to address the reviewers’ concerns, especially different fine-tuning strategies, to show that SCHaNe is promising. But the reviewers were not convinced.

After the rebuttal, R# 7hRi commented: if the motivation of this article is "...to address the issue of how cross-entropy can result in sub-optimal generalization rather than building a dedicated method for few-shot learning...." , the author should focus on more analysis and experiments instead of comparing with methods such as few-shot learning. R# owdS commented: In the experimental row for BEIT-3+SCL, the model solely utilizes supervised contrastive learning to fine-tune downstream classification tasks, which implies that it can yield results on the test set without classification weights.

The AC feels that the paper seemed to oversell on achieving SOTA using BEIT3 + SCHaNe, while didn’t show convincing results that the proposed method is indeed advancing general supervised contrastive learning at various setups. BEIT3 is a large-scale image-text pretraining system, and the proposed method seems to help BEIT3 on fine-turning. There is discrepancy between what the paper claims and what it truly delivers. Hence the AC recommends “reject”.

The authors contacted the AC for misconduct of a reviewer. After careful examination, the AC feels that the cause of the issue is simply typo. The AC didn’t see ill intention from the reviewer that would warrant further investigation.

**Justification For Why Not Higher Score:**

Two reviewers are negative. The only positive reviewer R# Snpe made the following comments after the rebuttal:

"However, I still have concerns regarding the novelty of the work. Thanks to Reviewer 7hRi, I've come to realize that the paper 'Supervised contrastive learning with hard negative samples' puts forth a pretty similar concept. I also believe that the manuscript could further benefit from additional experimentation.

To sum up, while I lean towards the acceptance of this paper, I would not dispute a decision to reject it."

Therefore the novelty and contribution were not recognized by the reviewers.

**Justification For Why Not Lower Score:**

N/A

---

### Decision · Program_Chairs · 2024-01-16

Reject